# Multi-Task Learning Model for Kazakh Query Understanding

**DOI:** 10.3390/s22249810

**Published:** 2022-12-14

**Authors:** Gulizada Haisa, Gulila Altenbek

**Affiliations:** 1College of Information Science and Engineering, Xinjiang University, Ürümqi 830017, China; 2The Base of Kazakh and Kirghiz Language of National Language Resource Monitoring, Research Center on Minority Languages, Ürümqi 830017, China; 3Xinjiang Laboratory of Multi-Language Information Technology, Ürümqi 830017, China

**Keywords:** query understanding, multi-task learning, named entity recognition, question classification, Kazakh

## Abstract

Query understanding (QU) plays a vital role in natural language processing, particularly in regard to question answering and dialogue systems. QU finds the named entity and query intent in users’ questions. Traditional pipeline approaches manage the two mentioned tasks, namely, the named entity recognition (NER) and the question classification (QC), separately. NER is seen as a sequence labeling task to predict a keyword, while QC is a semantic classification task to predict the user’s intent. Considering the correlation between these two tasks, training them together could be of benefit to both of them. Kazakh is a low-resource language with wealthy lexical and agglutinative characteristics. We argue that current QU techniques restrict the power of the word-level and sentence-level features of agglutinative languages, especially the stem, suffixes, POS, and gazetteers. This paper proposes a new multi-task learning model for query understanding (MTQU). The MTQU model is designed to establish direct connections for QC and NER tasks to help them promote each other mutually, while we also designed a multi-feature input layer that significantly influenced the model’s performance during training. In addition, we constructed new corpora for the Kazakh query understanding task, namely, the KQU. As a result, the MTQU model is simple and effective and obtains competitive results for the KQU.

## 1. Introduction

Question classification (QC) and named entity recognition (NER) are two of the main subtasks in the query understanding task, frequently occurring in other natural language processing tasks. In previous studies, QC and NER were often modeled separately, where the QC was a classification task while the NER was a sequence labeling task. The two tasks are the same as the slot-filling and intent identification in spoken language understanding. Due to the correlation between these two tasks, training them jointly could improve both.. Currently, the research results of many scholars have also proved that this multi-task method of joint question classification and named entity recognition can more accurately express the semantic representation of a query sentence [1,2]. In query understanding tasks, standard deep learning methods such as seq2seq [3] architecture or other recurrent neural networks (RNN) [4,5]-based models can be used effectively to capture the grammar structure in a question. However, complex morphology often makes such methods less effective in the agglutinative language scenario. Moreover, until recently, most works on query understanding have focused on several specific languages such as English, German, Chinese, and others.

From previous research, it can be seen that QC and NER are closely related, and solving QU tasks through multi-task joint models is of great help in improving the model performance. Multi-task learning uses parallel learning to improve the QU performance by correlating different tasks. A large number of experiments, such as in Falke et al. [6] and Broscheit et al. [7] have proved the effectiveness of the multi-task joint training method and the performance improvement on several benchmark datasets such as in Atis [8] and Snips [9].

Therefore, this paper proposes a multi-task learning model based on QC and NER, which completes two different tasks within a computational framework. We focus on the Kazakh QU task, where the corpora of the joint tasks training dataset are in the Kazakh language. For example, the input query, ‘Тыйаншан үлкен шатқалы қай жерге oрналасқан’(where is the Grand Canyon of Tianshan Mountains?), sampled from the KQU corpus, is shown in Table 1. In Table 1, the QC works on the sentence-level to indicate that the question type is about a scenic location, while the NER focuses on the words-level to figure out that the scenic name is the “Grand Canyon of Tianshan Mountains”.

The multi-task model presented in this article starts with using the language’s morphological features and syntactic features as the text word embedding representation, then uses BiLSTM to obtain context-dependent features and the attention mechanism to obtain the important information of terms. Finally, we designed specific network layers for QC and NER separately to accomplish two different tasks. The contributions of our paper are as follows:This paper proposes a deep multi-task learning model (MTQU) that can solve QC and NER tasks, and can learn the interaction mode between the QC and NER through the parameter-sharing mechanism provided by the multi-task learning framework.We demonstrate the importance of multi-feature embedding for QU in agglutinative languages.We construct a Kazakh query understanding corpus (KQU).The proposed QU learning model on the benchmark dataset is evaluated and the effectiveness is verified experimentally.

The remainder of this paper is organized as follows. The next section introduces the related work. Section 3 describes our proposed model. Section 4 illustrates the datasets. Section 5 presents our experiments, including the evaluation metrics, hyperparameter settings, experimental results, and analysis. Section 6 offers the conclusion and future work.

## 2. Related Work

Query understanding is a critical component in question-answering systems. QU typically involves identifying the user’s intent and extracting semantic constituents from the natural language query, two tasks that are referred to as question classification and named entity recognition. There are a significant number of QU tasks and approaches, and we briefly review the most widely-used methods in this paper.

### 2.1. Named Entity Recognition

NER can be treated as a sequence labeling task. Popular approaches to solving sequence labeling problems include conditional random fields (CRFs) [10] and RNN [11]. Most current literature can be categorized into the attention model [12] and the pre-training model [13]. In the past two years, researchers have proposed new network models combined with gazetteers [14], dynamic gazetteers [15] and multiple features [16], and they have also achieved the SOTA in other language research. However, deep-learning-based methods still need to improve. First, the deep learning methods require a large amount of training data to improve the accuracy of the NER. Secondly, entities have unclear boundaries, multiple nesting, and other problems that need to be analyzed. Finally, the labeled corpus can only cover some entities. In the Kazakh NER research, Haisa et al. [17] proposed a NER model for the tourism field, which takes stem features and a named entity gazetteers graph as the inputs, and obtains deep feature information with a gated-graph neural network with an attention mechanism. The effectiveness of the proposed model was verified on the Kazakh-named entity recognition dataset. It was also confirmed that the deep learning model fused with stemming features and gazetteers was better than the existing methods in improving the entity recognition’s accuracy and generalization ability.

### 2.2. Question Classification

QC can be treated as a semantic query classification problem, and popular classifiers such as support vector machines (SVMs) [18] and deep neural network methods can be applied.

Deep learning technologies are becoming a dominant method in text classification because they allow for the automatic learning of text features and are more generalized. Kim et al. [19] proposed a convolutional neural network sentence classification method based on word vectors. Dachapally et al. [20] extended conventional neural network (CNN) architecture that can first classify a question into a broader category and, based on prior knowledge, can then type it into a more specific category. Xia et al. [21] proposed attention-based long short-term memory (LSTM) architecture, which connects continuous hidden states of previous time steps to the current time step and applies an attention mechanism to these hidden states. Chotirat et al. [22] demonstrated that POS tags can improve the questions’ classification performance. In the Kazakh QC study, Haisa et al. [23] proposed a QC model that combined the morphological multi-language feature with the deep learning model and verified that the method solved the data sparsity problem better in the QC task of agglutinative languages such as Kazakh than when using only a deep learning framework.

### 2.3. Joint Model for NER and QC

The joint model for question classification and named entity recognition are the same as the slot-filling and intent identification in spoken language understanding. Such a joint model simplifies the query understanding process, as only one model needs to be trained and fine-tuned for the two tasks. Recently, RNNs and encoder–decoder neural network models [3] have been successfully applied in natural language understanding tasks. The attention mechanism introduced in the literature of [24] enables the encoder–decoder architecture to learn to align and decode simultaneously. Mrini et al. [25] proposed a multi-task learning method for QU with data augmentation in the field of medicine, first establishing an equivalent relationship between the two based on the summary of the problem and the definition of the intent recognition task. Then, they proposed a parameter sharing mechanism, specifically, a constraint for decoder parameters to be closed and gradually loosened as they moved to the highest layer. The experiments showed that the multi-task learning method was better at learning in the QU task than the single task.

Multi-task learning in Kazakh is still in the preliminary research stage. As far as we know, this article is the first attempt to study QU tasks based on multi-task learning models. In contrast, we consider both the word-level and sentence-level features with a deep neural network to identify the better quality in the query understanding tasks. Furthermore, we construct a Kazakh QU corpus. Thus, this study has tremendous research significance.

## 3. Methodology

### 3.1. Multi-Task Learning Model Structure

In our work, we focused on Kazakh query understanding. We utilized a deep learning model that integrated Kazakh linguistic characteristics in response to Kazakh linguistic features.

The multi-task learning model (MTQU), based on the QC and NER presented in this paper, were composed of five layers: a feature extraction layer, BiLSTM layer, attention layer, pooling layer and the output layer. Figure 1 shows a detailed model structure where, for instance, the sentence ‘Тыйаншан үлкен шатқалы қай жерге oрналасқан?’ is illustrated. In Figure 1, each word corresponds to one named entity label, and a specific question classification is assigned for the whole sentence.

### 3.2. Feature Extraction Layer

Kazakh is a typical agglutinative language, meaning that adding a prefix or suffix to the same root can generate hundreds or thousands of words. This feature is likely to cause data sparseness problems in natural language processing. To solve the data sparseness problem effectively, it is necessary to break the words down into stems and morphemes through a morphological analysis. To illustrate this, consider the following English phrase ‘People who are currently traveling’, which can be translated into Kazakh with only one word, ‘саяхаттағылардың’, which can then be broken down into the root and additional suffixes, ‘саяхат+та+ғы+лар+дың.’ Where the first section is the stem, the last four spliced behind it are suffixes, and these four suffixes are very special. There are two inflectional and two derivational suffixes. What is more interesting is that each time you add a suffix, the part of speech of the stem changes once. Therefore, you can see the complex morphological features of Kazakh through this example.

The feature representation layer maps each word to a high-dimensional vector space. The vector representation of the word, $w_{i}$, is $x_{i}\in R^{d_{w}}$, and by looking up the word embedding matrix, $E_{w}\in R^{d_{w}\times \left|V\right|}$ is calculated, where $d_{w}$ represents the dimension of the word embedding. This article uses pre-trained token vectors $w_{token}$ and stems vectors $w_{stem}$ as the fixed-size vectors for each word. Through the research of [17,23], it was found that in the Kazakh QC and NER study, lexical features such as stems and affixes effectively avoided data sparsity and improved the recognition accuracy; therefore, this study also used these two characteristics as its important input information. It has also been shown in several experiments and data analyses that syntactic features such as phrase markers, whether the current word starts a sentence, and whether the current word is Latin, can also enhance the model’s accuracy in identifying named entities and questions. Finally, based on previous research, this paper combines the morphological features, word-level features, and sentence-level feature as the final input of the model.

Tokens: divide the original text with spaces and punctuation marks as separators. Many natural language processing tasks use this feature.

Stem (root): obtained from previous research work. The stem and affix information were obtained through a morphological analysis system. Several agglutinative language processes use these features.

Suffixes: Kazakh, as like other agglutinative languages, has inflectional and derivational affixes. The main feature of these two types is that inflectional affixes very often only add a minute or delicate grammatical meaning to the stem and do not change the word class to which they attach. Derivational affixes often change the lexical meaning. The nominal suffix is also important in NER. There were 39 types of non-transitional suffixes and 4 types of transitional suffixes.

Gazetteers: these were obtained from the Kazakh NER [17] task by the researchers focused on the base of the Kazakh and Kirghiz languages at the national language resource monitoring and research center on monitory languages.

Phrases tagging: as mentioned above, we used an automated phrase tagging system to tag the token information. Two types of Kazakh phrases were used here: noun phrases and verbal phrases.

In this article, the rich features discussed above serve as the input layers for the neural network. The overall embeddings can be expressed as:(1)$$w_{i}=w_{i}^{token}⊕w_{i}^{stem}⊕w_{i}^{phrase}⊕w_{i}^{suffix}⊕w_{i}^{gazetteers}⊕w_{i}^{start}⊕w_{i}^{latin},$$
where ⨁ represents a concatenate operation for linking various feature vectors, $w_{i}^{token}$ is the token, $w_{i}^{stem}$ is the stem, $w_{i}^{suffix}$ is the suffixes, $w_{i}^{phrase}$ is the phrase feature,$\;w_{i}^{gazetteers}$ is the named entity dictionary, $w_{i}^{start}$ is the current word as the beginning of a sentence, and $w_{i}^{latin}$ is the current word as a Latin word.

### 3.3. LSTM Layer

LSTM (long short-term memory) has strong sequence modeling capabilities and can capture contextual information at a longer distance. LSTM controls the input and output information through three special gate structures. To obtain the sequential characteristics and context-dependent information about words, the model uses a weight-sharing mechanism at the BiLSTM layer to share the weight parameters of QC and NER tasks to improve the feature representation. Specifically, the word vector output by the feature representation layer $w_{1}$,$\;w_{2}\ldots w_{i}\ldots w_{n}$ uses the bidirectional LSTM model to generate a hidden vector sequence $h_{1}$,$\;h_{2}\ldots h_{i}\ldots h_{n}$, encodes the context word of the entire question S in $h_{i}$, and finally maps the $w_{i}$ to the context representation space:(2)$$h_{f}=\vec{LSTM}\left(w_{i},\beta \right),\;i\in \left[1,n\right],$$
(3)$$h_{b}=\overset{↼}{LSTM}\left(w_{i},\beta \right),\;i\in \left[1,n\right],$$
(4)$$h_{i}=\left\{h_{1}^{f}:h_{1}^{b},h_{2}^{f}:h_{2}^{b},\ldots h_{i}^{f}:h_{i}^{b}\ldots h_{n}^{f}:h_{n}^{b}\right\},$$
where $h_{f}$ is the forward hidden layer, $h_{b}$ is the backward hidden layer, β represents the model parameters of the LSTM, and $h_{i}$ is the output of the BiLSTM layer.

### 3.4. Attention Layer

In the question sentence, not all words are necessary to identify the named entities and intent classification; therefore, attention mechanisms are introduced to extract words important to the QU task and aggregate the importance representations of each word to obtain an attention representation. The attention weight matrix of each word is obtained through the attention mechanism, and the text sequence $C_{i}$ is obtained by combining the output of all hidden layers:(5)$$C_{i}=\sum_{j=1}^{n}a_{it}h_{it},$$
where, $h_{it}$ represents the LSTM hidden layer state of the encoder at the t-th time; n represents the length of the input sentence; $a_{ij}$ represents the attention distribution probability of the output at the t-th time, which is then calculated using the softmax function. The calculation formula is as follows:(6)$$a_{it}=\frac{exp\left(e_{it}\right)}{\sum_{k=1}^{n}exp\left(e_{it}\right)},$$
(7)$$e_{it}=tanh\left(V_{a}h_{it}+b_{a}\right),$$
where, $e_{it}$ represents the evaluation score of influence on i outputs at t moments; $V_{a}$ and $b_{a}$ are the weight matrices.

### 3.5. Pooling Layer

The QU model in this paper adds an averaging pooling operation after the attention layer, intended to improve the model’s generalization ability and avoid overfitting. The pooling layer extracts the most representative feature:(8)$$C_{max}=max\left(c_{1},c_{2},\ldots c_{i},\ldots c_{n}\right),$$

We used the max-pooling method. In the end, we spliced together all the pooling layers in order to prevent over-fitting and enhance the robustness of the model.

### 3.6. Output Layer

The output layer of the model feeds the results of the pooled layer $C_{max}$ into two different represents, namely, the $U_{QC}$ as a text representation representing the QC and the $U_{ENR}$ as a text representation representing the NER task.

This paper uses the softmax function for the text classification to obtain the final classification result. The final result of the QC task is predicted to be $P_{QC}$. The outcome of the NER task is predicted to be $P_{NER}$.

Compared with the classification problem, the current prediction label in the sequence labeling problem is related not only to the input feature of the current input but also to the previous prediction label; that is, there is a mutual dependence between the prediction labels. CRFs are a conditional probability distribution model of input and output random variables. Consequently, we add the CRF layer above the $P_{NER}$ to jointly decode the best chain of labels of the question. 

For the multi-task learning QU model, $L_{QC}$ and $L_{NER}$ are used as loss functions for QC and NER, respectively:(9)$$L_{QC}=-\frac{1}{N}\sum_{i}^{N}\log\left(P_{QC}\right),$$
(10)$$L_{NER}=-\frac{1}{N}\sum_{i}^{N}\log\left(P_{NER}\right),$$

This article combines the $L_{QC}$ and $L_{NER}$ as the final objective function of multi-task learning, and the final joint objective is formulated as:(11)$$L=\alpha L_{qc}+\beta L_{ner},$$
where α and β are tunable parameters that measure the impact of the two tasks.

## 4. Datasets

We constructed a query understanding corpus (KQU) of the tourism domain in Kazakh and the datasets were from popular tourism websites. In the first step, we collected the question from websites such as Ctrip, Qunar, Tuniu, and Baidu Encyclopedia. Then, the documents were divided into simple question sentences. After removing the duplicates, the questions were imported into the manual marking system for further proofreading and correction. Our corpus was annotated by six Kazakh native students whose mother tongue was also the Kazakh language. During the annotating process, we adopted a cross-labeling strategy. There were two groups of labelers and each group labeled part of the data, then exchanged data with another group and discussed ambiguities in the labeling.

In terms of the types of questions and entities, we used eight categories of entity tags and twenty-two question types to provide more comprehensive information on tourist sites and tourist activities. The entity types were Person, Location, Scenic, Specialty, Organization, Culture, Time, and Nationality. The question types were Attractions, Climate type, Price, Route, Address, Area, Comments, Foods, Traffic, People, Administrative region, Time, Distance, Custom culture, Shorthand, Phone number, Accommodation, Altitude, Attraction level, Nickname, Attraction types, and Popularity level. Furthermore, we also used the tourism gazetteers [17]. The datasets were divided into three sets in an 8:1:1 ratio: training set, validation set, and test set, respectively. Table 2 shows the information specific to the distribution of each language question.

## 5. Experimental Settings

This section describes the model evaluation measures, parameter settings, and baseline settings in the multi-task learning QU model.

### 5.1. Evaluation Measures

We used three evaluation metrics in the query understanding experiments. We used the F1-score (NER-F1) for the named entity recognition task, and the CoNLL-2003 evaluation script calculated the F1-score. We used accuracy (QC-Acc) for the question classification task. It was simply the fraction of the query with a correctly predicted user intent. Moreover, the sentence-level semantic frame accuracy (Sent-Acc) was used to indicate the general performance of both tasks, which referred to the proportion of the query whose NER tags and question labels were both correctly-predicted in the whole corpus.

### 5.2. Baselines

To verify the validity of the MTQU model, we mainly compared the two groups of recent natural language understanding models.

The first group of previous models was the two recently introduced RNN-based spoken language understanding models: the Attention-BiRNN [3] and BiLSTM-LSTM [4].

Attention-BiRNN [3]: used an encoder–decoder model containing two RNN models as an encoder for the input and a decoder for the output.

BiLSTM-LSTM [4]: used BLSTM-LSTM encoder–decoders with a focus mechanism for the spoken language understanding, slot-filling task.

The second group of previous models included the Kazakh named entity recognition [17] and the Kazakh question classification model [23].

WSGGA-NER [17]: in the Kazakh NER task, this used an attention-based gated graph neural network, integrated with token, stem and named entity gazetteers features.

Multi-QC [23]: in the Kazakh QC task, this used a convolution and gated recurrent neural network, integrated with morphological multi-features, such as the stem, suffix, POS and phrases.

### 5.3. Implementation Details

We implemented all classification models within the Pytorch learning tools. We trained the models for 50 epochs and selected the best performing model. ELMO was used in this study for the token and stem embedding training. The dimension of ELMO was 1024. The other input embeddings, such as POS, suffixes, gazetteers, phrases, start and Latin dimensions, were 300. Moreover, we employed a two-layer BiLSTM with a hidden size of 200 and a dropout rate of 0.5. We used the Adam optimizer with a learning rate of 0.001.

## 6. Results and Analysis

In this section, our new proposed MTQU model was trained and tested on our MQU corpus.

### 6.1. Comparison with Previous Approaches

Our first experiment was conducted on the KQU dataset and compared with the currently existing approaches by evaluating their question classification accuracy, named entity recognition and sentence accuracy. A detailed comparison is given in Table 3. The WSGGA-NER [17] and Multi-QC [23] models were designed for single-named entity recognition or a single-question classification task. Hence only the QC-Acc or F1 scores are given.

The first group of models consists of state-of-the-art joint intent classification and slot-filling models. A sequence-based joint model using an encoder–decoder focus model (BiLSTM-LSTM [4]) and encoder–decoder RNN with aligned inputs (i.e., Attention-BiRNN [3]), can solve a query understanding task and have comparable results. The WSGGA-NER [17] (only for NER) model uses stem feature and entity gazetteers in the input layer, then a fusion of all features by an attention-based gated graph network. The Multi-QC [23] (only for QC) model uses stem, suffix, POS, and phrase features in a hybrid network model. It is worth mentioning that all the recent RNN networks presented here used word embedding obtained from the word2vec language model. This has been introduced in the corresponding literature.

As shown in Table 3, the new proposed MTQU model outperformed the previous models’ results on three evaluation metrics. The BiLSTM-LSTM model outperformed the Attention-BiRNN model by a focus mechanism. It seems that the BiLSTM-LSTM model with a focus mechanism was more robust for sentence accuracy in our experiment. Additionally, the MTQU model outperformed the single-named entity recognition (WSGGA-NER) and question classification model (Multi-QC). This was probably due to the joint modeling and efficient multi-feature-based ELMO embedding in the multi-task learning structure, which may have sufficiently modeled the label structures.

### 6.2. Multi-Task Model vs. Individual Models

We used the MTQU for both the named entity recognition and question classification tasks. We performed a comparative study of the performance of this model with the individual models, i.e., the tasks of the question classification and named entity recognition were built in isolation. Then, the results could be compared to verify the effectiveness of the shared training parameters. Table 4 demonstrates the multi-task and individual model’s results.

The multi-task part of the model performed superior to the individual models for both tasks. Our multi-task model learned the correlation between the two tasks and this shared information provided helpful evidence for both tasks. The experimental results prove that the multi-task model performed better in all the settings.

### 6.3. Ablation Analysis

This set of experiments aimed to compare with the standard MTQU model and demonstrate the advantage of performing different features and multi-task modeling within the proposed MTQU architecture. To clarify our point, we categorized the multi-language features into word-level and sentence-based features. The details can be found in Table 5.

For the KQU dataset, we conducted an ablation study on the KQU. In Table 6, we can observe that it yielded significant performance gains for all the methods, and the word-level and sentence–level features.

We compared the MTQU model with different features. Without the word-level features, the sentence accuracy dropped to 81.76% (from 83.58%). We further compared the MTQU model without sentence level features and the sentence accuracy dropped to 82.05% (from 83.58%), as in Table 6, showing how the multi-task learning QU model, by exploiting the language representation power of multi-features, improved the generalization capability.

## 7. Conclusions

This paper proposes a multi-task learning query understanding model in the Kazakh tourism domain, aiming at addressing the poor generalization capability of the traditional query understanding task in a complex morphological language. The experimental results show that our proposed joint question classification and named entity recognition models model the efficacy of exploiting the relationship between the tasks. Moreover, because the Kazakh language has complex morphological features, a neural network model integrates the multi-features of linguistics. The input layer obtains the contextual information using word-level and sentence-level features. The BiLSTM captures the context in longer sentences, and the subset task layer obtains the best representation results. Our MTQU model achieves a significant improvement in the question classification accuracy, named entity recognition F1, and sentence-level accuracy for KQU datasets over the previous state-of-the-art models. In the next step, first, we will expand the corpus of Kazakh QU, then, we will design a new neural network structure to improve the accuracy of the Kazakh question-answering systems.

## Figures and Tables

**Figure 1 sensors-22-09810-f001:**
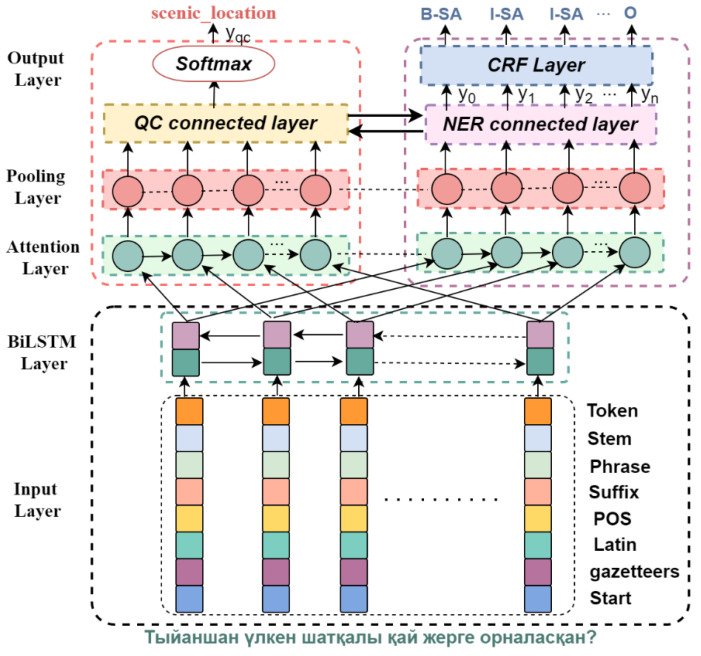
Illustration of the architecture of the MTQU model. (Kazakh can be written using both right-to-left (Arabic) and left-to-right (Latin or Cyrillic) scripts, and this work uses Cyrillic scripts).

**Table 1 sensors-22-09810-t001:** An example sentence from KQU corpus.

**Query**	Тыйаншан	лкен	шатқалы	қай	жерге	oрналасқан	?
**NER_tags**	B-SA	I-SA	I-SA	O	O	O	O
**QC_label**	Scenic_location						

**Table 2 sensors-22-09810-t002:** Datasets and statistics of KQU.

Types	Size
train	5600
dev	700
test	700
Question types	22
NER types	8

**Table 3 sensors-22-09810-t003:** Comparison with published approaches.

Model	QC-Acc	NER-F1	Sent-Acc
Attention-BiRNN [3]	87.81	88.65	79.76
BiLSTM-LSTM [4]	88.25	88.33	80.02
WSGGA-NER [17]	--	89.61	--
Multi-QC [23]	88.86	--	--
MTQU (Ours)	92.28	91.73	83.58

**Table 4 sensors-22-09810-t004:** Multi-task Model vs. Individual Models.

Model	QC-Acc	NER-F1
MTQU (Only QC)	90.89	--
MTQU (Only NER)	--	90.61
MTQU (Multi-task model)	92.28	91.73

**Table 5 sensors-22-09810-t005:** Multi features of the Kazakh language.

Word-Level Features	Sentence-Level Features
Stem	Gazetteers
First-suffix	Noun phrase tagging
Second-suffix	Verb phrase tagging
Third-suffix	Start of the sentence
Nominal-suffix	
Latin words	

**Table 6 sensors-22-09810-t006:** Performance with different features.

Model	QC-Acc	NER-F1	Sent-Acc
Contacted all-features	92.28	91.73	83.58
Remove word-level features	90.39	90.68	81.76
Remove sentence-level features	90.75	89.87	82.05

## Data Availability

Not applicable.

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
