# Peer review of "Multi-Task Learning Model for Kazakh Query Understanding"

_sensors, 2022, doi:10.3390/s22249810_

Round 1

Reviewer 1 Report

The paper is devoted to many aspects that are associated with deep machine learning technologies for solving problems that are associated with automatic processing and understanding of natural language. The authors of the paper describe the main problems that arise during natural language processing. So, problems can be during the extraction of relationships, information retrieval, machine translation, analysis of other aspects. Therefore, the authors of the paper decided to pay attention to Question Classification (QC) and Named Entity Recognition (NER). The authors of the paper propose a new multi-task learning model for query understanding (MTQU) based on QC and NER. In general, this research can be in line with the current level of global scientific community output. Testing and comparative analysis of the results of the study were carried out on the KQU corpus. This corpus is described in detail by the authors in this paper. From the description it is clear that the corpus consists of texts in the Kazakh language, and the topic is tourism. In addition, the main problems that may arise during the automatic understanding of the Kazakh language are clearly described. On the basis of comparative analysis, it can be argued that qualitative indicators still depend on the input data.

However, in my humble opinion, the paper is not free from a number of small flaws that need to be addressed. 1) First of all, it is striking that there are no references to previous works of the world scientific community (2021-22), which are constantly presented at conferences focused on working with text modality or connected to it through multimodal data processing (EMNLP, ACL, NAACL, INTERSPEECH, EUSIPCO, ICASSP, SPECOM and others) or collection of hulls (LREC and others), therefore it is recommended to expand the section describing previous works, including modern solutions. 2) Is it correct to assume that the assembled corpus will not be open-source to the scientific community? It's not understood from the paper. 3) It is not entirely clear whether the results of the metrics are compared with existing approaches that were also implemented by the authors and trained on the assembled corpus or not? 4) Finally, the style of the paper requires minor revision due to the presence of spelling and punctuation errors.

The rest of the paper is interesting, but only the moments that are presented in the form of current shortcomings confuse. It seems to me that all the proposed additions will only improve this paper, and it will be useful and interesting to many specialists who associate their research with automatic natural language processing, but only after completion.

Author Response

Response to Reviewer 1 Comments

Point 1: First of all, it is striking that there are no references to previous works of the world scientific community (2021-22), which are constantly presented at conferences focused on working with text modality or connected to it through multimodal data processing (EMNLP, ACL, NAACL, INTERSPEECH, EUSIPCO, ICASSP, SPECOM and others) or collection of hulls (LREC and others), therefore it is recommended to expand the section describing previous works, including modern solutions.

 Response 1: The references (2021-EMNLP, 2022-ACL, 2022-NAACL) mentioned in the comments have been added. These references are in the manuscript as [6], [7] and [15].

Point 2: Is it correct to assume that the assembled corpus will not be open-source to the scientific community? It's not understood from the paper.

Response 2: We decided to make all the data public in the future after further improving the corpus.

Point 3: It is not entirely clear whether the results of the metrics are compared with existing approaches that were also implemented by the authors and trained on the assembled corpus or not?

Response 3: To further verify the validity of the MTQU model, a comparison experiment between the multi-task model and the individual model has been added to the revision (Section 6.2).

Point 4: Finally, the style of the paper requires minor revision due to the presence of spelling and punctuation errors.

Response 4: we are very sorry for our incorrect writing. First, we carefully analyzed and revised the manuscript, and then the manuscript was checked by a native English-speaking colleague.

Reviewer 2 Report

Based on query understanding, this paper proposes a multi-task learning model based on named entity recognition and question classification, and constructs a new data set to adapt to this task, and compares the relevant experimental methods. Overall, the model is well designed, especially for the agglutination and complex morphology of the Kazakh language, the use of multi-feature representation, and BiLSTM model for feature learning. It is suggested that in the ablation experiment, the multi-task part of the model can be separated into two tasks for training and prediction, and then the results can be compared to verify the effectiveness of shared training parameters. In addition, the font of some symbols in the article needs to be modified.

Author Response

Response to Reviewer 2 Comments

Point 1: It is suggested that in the ablation experiment, the multi-task part of the model can be separated into two tasks for training and prediction, and then the results can be compared to verify the effectiveness of shared training parameters.

 Response 1: To further verify the validity of the MTQU model, a comparison experiment between the multi-task model and the individual model has been added to the revision (Section 6.2).

Point 2: In addition, the font of some symbols in the article needs to be modified.

Response 2: we are very sorry for our incorrect writing. First, we carefully analyzed and revised the manuscript, and then the manuscript was checked by a native English-speaking colleague.

Round 2

Reviewer 1 Report

The authors of the paper finalized the paper and made all the necessary changes. In this form, the paper can be useful and interesting for many specialists who associate their research with automatic natural language processing. In general, the paper can be accepted for publication.